# Nucleotide-Binding Leucine-Rich Repeat Genes *CsRSF1* and *CsRSF2* Are Positive Modulators in the *Cucumis sativus* Defense Response to *Sphaerotheca fuliginea*

**DOI:** 10.3390/ijms22083986

**Published:** 2021-04-13

**Authors:** Xue Wang, Qiumin Chen, Jingnan Huang, Xiangnan Meng, Na Cui, Yang Yu, Haiyan Fan

**Affiliations:** 1College of Bioscience and Biotechnology, Shenyang Agricultural University, Shenyang 110866, China; wxue9507@163.com (X.W.); qiuminchen2019@163.com (Q.C.); 15037943009@163.com (J.H.); 2019500018@syau.edu.cn (X.M.); cuina@syau.edu.cn (N.C.); 2College of Horticulture, Shenyang Agricultural University, Shenyang 110866, China; 3Key Laboratory of Protected Horticulture of Ministry of Education, Shenyang Agricultural University, Shenyang 110866, China

**Keywords:** *Cucumis sativus*, NBS-LRR, expression analysis, *Sphaerotheca fuliginea*, transient genetic transformation

## Abstract

Cucumber powdery mildew caused by *Sphaerotheca fuliginea* is a leaf disease that seriously affects cucumber’s yield and quality. This study aimed to report two nucleotide-binding site-leucine-rich repeats (NBS-LRR) genes *CsRSF1* and *CsRSF2*, which participated in regulating the resistance of cucumber to *S. fuliginea*. The subcellular localization showed that the CsRSF1 protein was localized in the nucleus, cytoplasm, and cell membrane, while the CsRSF2 protein was localized in the cell membrane and cytoplasm. In addition, the transcript levels of *CsRSF1* and *CsRSF2* were different between resistant and susceptible cultivars after treatment with exogenous substances, such as abscisic acid (ABA), methyl jasmonate (MeJA), salicylic acid (SA), ethephon (ETH), gibberellin (GA) and hydrogen peroxide (H_2_O_2_). The expression analysis showed that the transcript levels of *CsRSF1* and *CsRSF2* were correlated with plant defense response against *S. fuliginea*. Moreover, the silencing of *CsRSF1* and *CsRSF2* impaired host resistance to *S. fuliginea*, but *CsRSF1* and *CsRSF2* overexpression improved resistance to *S. fuliginea* in cucumber. These results showed that *CsRSF1* and *CsRSF2* genes positively contributed to the resistance of cucumber to *S. fuliginea*. At the same time, *CsRSF1* and *CsRSF2* genes could also regulate the expression of defense-related genes. The findings of this study might help enhance the resistance of cucumber to *S. fuliginea*.

## 1. Introduction

Cucumber (*Cucumis sativus*) is one of the largest protected vegetable cultivation in China. Cucumber powdery mildew caused by *S. fuliginea* is an important disease of cucumber that seriously affects cucumber’s yield and quality. Cucumber powdery mildew has some characteristics, such as wide distribution, short incubation period and so on [1]. It reduces cucumber yield by 20%–50%, or even 0% [2]. At present, planting resistant cultivars is the most important and effective way of controlling cucumber powdery mildew. It is vital to clone key genes related to the infection of *S. fuliginea* and explore its mechanism of action so as to improve the resistance of cucumber to *S. fuliginea*.

Producing toxins to destroy the host’s defense system and make plants susceptible to diseases is one way in which pathogens infect plants [3]. To control pathogens, plants have formed an innate immune system [4,5]. This immune system has two layers: the first layer is called pathogen-associated molecular patterns (PAMPs)-triggered immunity (PTI), which is the recognition of PAMPs by pattern-recognition receptors on the cell surface; the second layer is called effector-triggered immunity (ETI), which is the recognition of pathogenic factors within the bacteria. This process mainly triggers a hypersensitive response (HR), which causes programmed cell death to block pathogens from absorbing nutrients and limit the proliferation and spread of hyphae [6]. Furthermore, HR can activate ethylene (ET), SA, jasmonic acid (JA), and other defense signaling pathways [7,8]. Plant disease resistance proteins (R proteins) are vital in the ETI [9]. Nucleotide-binding site-leucine-rich repeat (NBS-LRR) proteins are the main types of R proteins.

NBS-LRR proteins are classified into two subfamilies based on the motifs located in the N-terminal region [10]: TIR-NBS-LRR (TNL) and non-TIR-NBS-LRR (CC-NBS-LRR (CNL)) [11]. These proteins have three domains: nucleotide-binding site (NBS) (nucleotide-binding and ATPase (NB-ARC)) domain, leucine-rich repeats (LRR) domain, and Toll-interleukin receptor (TIR)/coiled-coil (CC) domain. The NBS domain can change ADP to ATP so that it can stimulate downstream HR [12]. It can also participate in all kinds of processes, such as cell growth, differentiation, cytoskeletal organization, vesicle transport, and defense. The LRR domain is important in the protein–protein interaction [13]. The TIR domain is involved in resistance specificity and signaling [14]. The CC domain is the key to induce an immune response by interacting with downstream signal components [15].

The NBS-LRR protein senses pathogen effectors and causes them to trigger an immune response of plants to develop a defense response against pathogens, such as bacteria, viruses, and fungi [16]. Genes encoding the NBS-LRR protein in different plants can confer resistance to plants directly or indirectly, thus helping plants resist attack by pathogens [17,18]. For example, the NBS-LRR gene *Pm60* confers resistance to powdery mildew in wheat [19]; the NBS-LRR gene *SacMi* participates in plant resistance to *Meloidogyne* spp. in wild eggplant [20]; and the tomato NBS-LRR gene expression has a positive correlation with plant resistance to *Phytophthora infestans* [21].

A complex network of different signal transductions exists in the interaction between plants and pathogens [22]. The NBS-LRR protein can directly or indirectly recognize and interact with avirulence (Avr) protein products [23]. This recognition of effectors can change the NBS-LRR protein from an inhibited state to an activated state, thereby activating downstream signal transduction to generate a defense response. The barley CNL protein MLA10 can interact with WRKY (transcription repressor) and MYB6 (transcription activator) to activate the defense response [24,25]. The rice CNL protein Pb1 interacts with OsWRKY45 and stabilizes it to develop resistance to *Magnaporthe oryzae* [26]. The *Arabidopsis* TNL protein RPS4 can recognize AvrRps4 to activate HR [27,28,29]. The potato CNL protein Rx recognizes the potato virus coat protein (PVX CP) and interacts with the cytoplasmic activation protein RanGAP2 to activate the signaling pathway [30].

The NBS-LRR function in *S. fuliginea* infection and the defense mechanisms have not been described in cucumber. Wan et al. performed a genome-wide analysis of NBS-encoding genes within the whole cucumber genome. Moreover, they found two proteins, Cucsa.102240 and Cucsa.123410, which were predicted to have a domain with a sequence similar to the resistance to powdery mildew 8 (RPW8) *Arabidopsis* powdery mildew resistance gene family [31,32]. In addition, we used bioinformatics software to analyze these two proteins and found they have RPW8 domains. Therefore, they may be involved in the interaction between cucumber and *S. fuliginea* or may contribute to defense responses against *S. fuliginea*. They were named *CsRSF1* and *CsRSF2*. This study aimed to report whether two NBS-LRR genes *CsRSF1* and *CsRSF2*, participated in regulating the resistance of cucumber to *S. fuliginea*. In this study, *CsRSF1* (Cucsa.102240) and *CsRSF2* (Cucsa.123410) were cloned from B21-a-2-2-2 (highly susceptible to *S. fuliginea*) and B21-a-2-1-2 (highly resistant to *S. fuliginea*); their sequences showed no differences between the resistant and susceptible cultivars. They were typical NBS-LRR proteins as detected by the bioinformatics analysis. *CsRSF1* and *CsRSF2* were induced upon *S. fuliginea* infection, but their expression patterns were different. Subcellular localization analysis showed that CsRSF1 and CsRSF2 proteins were located in the plasma membrane and cytoplasm, whereas the CsRSF1 protein was also located in the nucleus. These genes responded differently to exogenous substances. The defense role dissections of *CsRSF1*/*CsRSF2*-silencing and *CsRSF1*/*CsRSF2*-overexpressing cucumber plants indicated that *CsRSF1*/*CsRSF2*, acting as positive regulators, were required for host resistance response to *S. fuliginea*.

## 2. Results

### 2.1. Gene Cloning and Sequence Analysis of CsRSF1 and CsRSF2

According to the sequence of the target gene found in the Phytozome database (https://phytozome.jgi.doe.gov/pz/portal.html, last accessed on 9 April 2021), the full-length CDS sequences of *CsRSF1*/*CsRSF2* were cloned from the experimental materials of B21-a-2-1-2 and B21-a-2-2-2. The comparison of the sequences showed no difference in CDS sequences of *CsRSF1*/*CsRSF2* between the B21-a-2-1-2 variety and B21-a-2-2-2 variety. The full CDS sequence of *CsRSF1* was 2457 bp, which encoded 818 amino acid (AA) residues with the molecular weight of 93.59 kDa and the theoretical iso-electric point (pI) of 6.05 (Appendix A). The full CDS sequence of *CsRSF2* was 2466 bp, which encoded 821 AA residues with a molecular weight of 94.25 kDa and a theoretical pI of 6.58 (Appendix A). In addition, the chromosomal distribution showed that *CsRSF1* was located on chromosome 6 (scaffold00927), and *CsRSF2* was located on chromosome 4 (scaffold01000) (Appendix A).

The analysis of the protein sequence showed that CsRSF1 contained an N-terminal RPW8 domain, an NB-ARC domain, and an LRR domain at the C-terminus (Appendix A). Similarly, CsRSF2 also had an N-terminal RPW8 domain, an NB-ARC domain, and an LRR domain at the C-terminus (Appendix A).

### 2.2. Subcellular Localization and Expression Patterns of CsRSF1 and CsRSF2

Bioinformatics prediction indicated that the CsRSF1 protein was localized in the cytoplasm and cell membrane, and the CsRSF2 protein was localized in the cytoplasm. To confirm this, the p35S:GFP-*CsRSF1*/*CsRSF2* fusion vector was constructed (Figure 1A), and the p35S:GFP-*CsRSF1*/*CsRSF2* and the control p35S:GFP vectors were separately introduced into tobacco leaves. Confocal microscopic examination showed that CsRSF1 and CsRSF2 proteins were both located in the cytoplasm and cell membrane. In addition, the CsRSF1 protein was also located in the nucleus (Figure 1B).

The expression patterns of *CsRSF1* and *CsRSF2* in different organs (leaves, cotyledons, stems and roots) of susceptible and resistant cultivars were analyzed by qRT–PCR (Figure 2). The results showed the expression patterns of *CsRSF1* and *CsRSF2* were different in various organs in the resistant and susceptible cultivars. Among the resistant cultivars, *CsRSF1* exhibited the highest expression in roots, followed by leaves and cotyledons, but it was weakly expressed in stems (Figure 2A). Nevertheless, *CsRSF1* in the susceptible cultivar was relatively highly expressed in leaves, followed by stems, roots, and cotyledons (Figure 2B). Moreover, *CsRSF2* in the resistant cultivars was significantly expressed in cotyledons but barely expressed in stems, leaves, and roots (Figure 2C). In the susceptible cultivars, *CsRSF2* was significantly expressed in cotyledons and stems, followed by leaves and roots (Figure 2D). This result indicated *CsRSF1* and *CsRSF2* genes showed varied expression patterns among organs in cucumber. The resistance of *CsRSF1* and *CsRSF2* may be different in various organs in the resistant and susceptible plants.

### 2.3. Response of CsRSF1 and CsRSF2 to Exogenous Substances

The transcript levels of *CsRSF1* and *CsRSF2* were determined in the susceptible and resistant cultivars at the two-leaf-one-heart stage after exogenous treatment with ABA, MeJA, SA, ETH, GA, and H_2_O_2_ to investigate the expression patterns of *CsRSF1* and *CsRSF2* genes in response to exogenous substances. The result indicated that the regulation of *CsRSF1* and *CsRSF2* was different between resistant and susceptible varieties of cucumber after exogenous treatment with MeJA, SA, ETH, and H_2_O_2_. In addition, *CsRSF1* and *CsRSF2* may be involved in phytohormone-related signaling pathways (Figure 3). Compared with the control plants, the levels of *CsRSF1* transcripts showed an upward trend at first and then a downward trend, which reached the highest level 12 h after ABA treatment in the resistant and susceptible cultivars (Figure 3A,B). The levels of *CsRSF2* transcripts decreased 12 h, 24 h, and 48 h after GA treatment in the resistant and susceptible cultivars (Figure 3C,D). Therefore, the results indicated that *CsRSF1* might be involved in the ABA signaling pathway, while *CsRSF2* might be involved in the GA signaling pathway.

### 2.4. Expression Patterns of CsRSF1 and CsRSF2 in the Resistant/Susceptible Cucumber Cultivars after S. fuliginea Inoculation

qRT–PCR was used to detect the transcription levels of *CsRSF1* and *CsRSF2* in the susceptible and resistant cultivars after inoculation with *S. fuliginea* (Figure 4). The comparison of the two cultivars showed that *CsRSF1* expression was induced at 6 h and peaked at 12 h in the susceptible cultivar. In the resistant cultivar, *CsRSF1* expression was largely induced at 3 and 6 h. The *CsRSF1* expression in the resistant cultivars were higher than that in the susceptible cultivars between 0 h and 9 h (Figure 4A). However, the *CsRSF2* expression reached a peak at 3 h in the susceptible cultivar. *CsRSF2* expression increased in the initial phase and reached a peak at 6 h in the resistant cultivar (Figure 4B). This finding suggested that *CsRSF1* and *CsRSF2* were induced upon *S. fuliginea* infection in the two cultivars.

### 2.5. Silencing of CsRSF1 and CsRSF2 Impairs Host Resistance to S. fuliginea

TRV-based virus-induced gene silencing (VIGS) was used to knock down *CsRSF1*/*CsRSF2* transcripts in Xintaimici to explore whether *CsRSF1*/*CsRSF2* was required for resistance of cucumber to *S. fuliginea* (Figure 5). A 3′-terminal fragment specific in *CsRSF1* and a 5′-terminal fragment specific in *CsRSF2* were inserted in the antisense orientation into the vector pTRV2, and the pTRV2:*CsRSF1*/*CsRSF2* recombinant plasmids were generated (Figure 5A). Subsequently, TRV:*CsRSF1*/*CsRSF2* viruses were injected into cucumber cotyledons. After inoculation with TRV:*CsRSF1* or TRV: 00 (as a control) viruses for 5 days and TRV:*CsRSF2* or TRV: 00 (as a control) viruses for 7 days, the mild chlorotic mosaic symptoms of TRV appeared in the cotyledons of the infected Xintaimici plants (Figure 5B). The TRV: 00 infections did not significantly affect the expression of *CsRSF1/CsRSF2* in Xintaimici plants. Meanwhile, the transcript level of *CsRSF1/CsRSF2* was markedly reduced 0.5–0.75/0.2–0.5 in TRV:*CsRSF1*/*CsRSF2*-infected (*CsRSF1/CsRSF2*-silencing) plants (Figure 5C). These results indicated that *CsRSF1* and *CsRSF2* were successfully transiently silenced in cucumber cotyledons.

Further, the cotyledons of TRV-infected plants were inoculated with *S. fuliginea* to evaluate the defense role of *CsRSF1/CsRSF2* (Figure 6). At 7 dpi with *S. fuliginea*, large areas of white powder (a symptom of powdery mildew disease) appeared on the cotyledons of *CsRSF1*/*CsRSF2*-silencing plants, while the areas of white powder in control and TRV: 00-treated plants (Figure 6A,B) were smaller. Moreover, many hyphae and branch spores invaded *CsRSF1*/*CsRSF2*-silencing plants, but only a few initial germ tubes and appressoria were observed in control and TRV: 00-injected cucumbers under the microscope (Figure 6A,B). In addition, the DI of the *CsRSF1*-silencing plants was 32.41; the DI of the control and TRV: 00-injected cucumbers were 22.22 and 24.44, respectively (Table 1); the DI of the *CsRSF2*-silencing plants was 36.11; and the DI of the control and TRV: 00-injected cucumbers was 16.68 and 17.77, respectively (Table 2). These results indicated that the silencing of *CsRSF1*/*CsRSF2* impaired resistance to *S. fuliginea*, and *CsRSF1* and *CsRSF2* were required for the cucumber defense response to *S. fuliginea* infection.

### 2.6. CsRSF1 and CsRSF2 Overexpression Improved Resistance to S. fuliginea in Cucumber

*CsRSF1*/*CsRSF2* was cloned into a pRI101-AN-GFP vector, and a GFP:*CsRSF1*/*CsRSF2* plasmid was generated to further investigate whether *CsRSF1*/*CsRSF2* was vital in cucumber disease resistance (Figure 7). The PCR analysis showed that the introduced genes could be detected in transient overexpression plants by chimeric primers between the *CsRSF1*/*CsRSF2* and GFP fusion genes (Figure 7A). Protoplast location experiments showed that the GFP signals were detected in the *CsRSF1*/*CsRSF2*–GFP co-expression regions, confirming the success of the transient overexpression of *CsRSF1*/*CsRSF2* in vivo (Figure 7B). In addition, the qRT–PCR assay demonstrated that the expression of the *CsRSF1*/*CsRSF2* genes increased by 3–5.5/20–23 compared with that in the GFP: 00-injected cucumber cotyledons (Figure 7C). These results indicated that *CsRSF1* and *CsRSF2* were transiently overexpressed in cucumber cotyledons successfully.

After inoculation with *S. fuliginea*, the *CsRSF1*/*CsRSF2*-overexpressing plants exhibited significantly enhanced resistance to *S. fuliginea* compared with the control and GFP: 00-injected plants (Figure 8). At 7 dpi with *S. fuliginea*, large areas of white powder were present on the cotyledons of the control and GFP: 00-treated plants, while the areas of white powder in *CsRSF1*/*CsRSF2*-overexpressing plants were smaller (Figure 8A). Furthermore, microscopic observation indicated that the hyphae abundance of *S. fuliginea* was low on the inoculated base sheaths of the *CsRSF1*/*CsRSF2*-overexpressing plants than on the sheaths of the control and GFP: 00-injected plants (Figure 8B). The finding provided supporting evidence that *CsRSF1*/*CsRSF2* overexpression increased resistance to hyphae development in *S. fuliginea*. The average disease indexes of the *CsRSF1*/*CsRSF2*-overexpressing plants infected with *S. fuliginea* were 24.88 and 17.33, whereas those of control and GFP: 00-injected plants were 34.59 and 32.00 (Table 3). These results indicated that *CsRSF1* and *CsRSF2* positively regulated cucumber resistance to powdery mildew caused by *S. fuliginea*.

### 2.7. CsRSF1 and CsRSF2 Modulated Expression Levels of Defense-Related Genes

*Chitinase* (HM015248), *CuPi1* (U93586.1), and *PR-1a* (AF475286) are three well-known *S. fuliginea*–related defense genes [33,34,35,36]. The expression patterns of the aforementioned three defense-related genes were analyzed in *CsRSF1*/*CsRSF2*-overexpressing and *CsRSF1*/*CsRSF2*-silenced cucumber plants as well as the control plants to examine whether *CsRSF1* and *CsRSF2* regulated these defense-related genes (Figure 9). The results showed that the transcriptional levels of *CuPi1* and *PR-1a* decreased in *CsRSF1*-silenced plants compared with the TRV: 00-infected control plants. Simultaneously, the transcriptional level of *Chitinase* significantly increased, whereas the transcriptional levels of *Chitinase* and *PR-1a* increased in *CsRSF1*-overexpressing plants compared with GFP: 00, while the transcriptional level of *CuPi1* decreased. However, the transcriptional levels of *CuPi1* and *PR-1a* in *CsRSF2*-silenced plants were significantly higher than those in the TRV: 00-infected control plants. Only the transcriptional level of *Chitinase* decreased, whereas the transcriptional levels of *Chitinase*, *CuPi1*, and *PR-1a* significantly increased in the *CsRSF2*-overexpressing plants compared with GFP: 00. The results indicated that the *CsRSF1*/*CsRSF2*-induced defense pathway might be related to different defense-related genes.

## 3. Discussion

Powdery mildew is a common and widespread plant disease of considerable agronomic relevance. Powdery mildew outbreaks often result in severe harvest losses [37]. Cucumber powdery mildew resistance is a particularly genetically complex trait, which often involves multiple genes, but the genetic mechanisms of cucumber resistance are still poorly understood. Recently, 67 NBS-LRR resistance gene homologs have been predicted in cucumber, some of which may be related to the resistance of powdery mildew [38]. Zhang et al. [39] found four QTLs of cucumber powdery mildew, which are *pm5.1*, *pm5.2*, *pm5.3*, *pm6.1*. The *pm5.2* locus has the highest contribution rate, and four NBS resistance genes (*BGICucGB009775*, *Csa009587*, *Csa009605*, *Csa009602*) were predicted in this region.

NBS-LRR is a class of proteins important in pathogen recognition and defense response signal transduction [40,41]. However, the role of NBS-LRR gene resistance to powdery mildew, which has been studied less and is mainly concentrated in wheat. For example, the wheat NBS-LRR gene *Pm21* has broad-spectrum resistance to wheat powdery mildew [42]. However, in cucumber–*S. fuliginea* interactions, the role of the NBS-LRR gene is unclear. In this study, CDS regions of *CsRSF1* and *CsRSF2* were cloned from the cucumber leaves and their functions between cucumber–*S. fuliginea* interactions were identified. The expression of *CsRSF1* showed a trend of increase at first and then decreased in resistant and susceptible cultivars challenged with *S. fuliginea*. The expression of *CsRSF2* was different in resistant and susceptible cultivars challenged with *S. fuliginea*. In the early stage of *S. fuliginea* infection, the expression of *CsRSF1* and *CsRSF2* was higher in resistant cultivars than in susceptible cultivars. These results indicated that both *CsRSF1* and *CsRSF2* were involved in cucumber response to *S. fuliginea* infections, having a positive regulatory role.

The subcellular localization of a protein is strongly related to its biological function in plants. The NBS-LRR protein, as important intracellular receptors, can directly or indirectly recognize and interact with Avr protein products inside the cell to generate a defense response [23]. The barley CNL protein MLA10 can interact with WRKY (transcription repressor) and MYB6 (transcription activator) to activate the defense response in the nucleus [24,25]. Similarly, *Arabidopsis* CNL proteins RPS5, RPS2 and RPM1 recognize the pathogen effectors and interact with protein kinases and activators to generate defense response in the plasma membrane [43,44]. Therefore, the subcellular localization of NBS-LRR proteins is especially important in the plant’s innate immune system. In this study, the subcellular localization of 35S::CsRSF1-GFP and 35S::CsRSF2-GFP fusion proteins in tobacco epidermal cells showed that both CsRSF1 and CsRSF2 were localized in the cytoplasm and plasma membrane; mean-while, CsRSF1 was also localized in the nucleus. CsRSF1 and CsRSF2 had potential functions in various parts of the cell.

Many studies verify a gene’s function by using silencing and overexpressing plants. Research showed that the tomato NBS-LRR gene *SLNLC1* silences attenuated tomato resistance to *Stemphylium lycopersici* [45]. The overexpression and silencing of the NBS-LRR gene *CaRGA* in chickpeas resulted in increased resistance and sensitivity to *Fusarium* wilt [46]. Currently, the application of stable transgenic technology to cucumber is still difficult. Hence, an experimental method was used for the transient agroinfiltration of cucumber cotyledons to obtain transient *CsRSF1* and *CsRSF2* silencing and overexpression plants [47]. In this study, inoculation with *S. fuliginea* resulted in the typical whitish appearance of powdery mildew disease on *CsRSF1* and *CsRSF2* silencing plants and also caused them to exhibit greater hyphal abundances and a lower DI than control plants. This result is contrary to *CsRSF1* and *CsRSF2* overexpression plants. The functional analysis showed that *CsRSF1* and *CsRSF2* overexpression significantly increased the resistance of cucumber to *S. fuliginea*, while *CsRSF1* and *CsRSF2* silencing significantly compromised resistance to *S. fuliginea*.

Different regulatory proteins might activate different kinds of genes in cucumber defense response to *S. fuliginea* [34]. *Chitinase* has a positive regulatory effect on the resistance of cucumber to *S. fuliginea*. SA signaling pathway induces PR gene expression, which can enhance resistance to a broad range of pathogens [48]. Therefore, PR gene expression is a sign of plant disease resistance. *PR-1a* is one of the PR genes. In this study, the expression of *PR-1a* was lower on *CsRSF1* silencing plants than control plants, higher on *CsRSF1* overexpression plants than control plants. The expression of *Chitinase* was lower on *CsRSF2* silencing plants than control plants, higher on *CsRSF2* overexpression plants than control plants. *CsRSF1* positively modulated the expression of *PR-1a*, while *CsRSF2* positively modulated the expression of *Chitinase*.

Plant hormones are essential for plants to respond to biological and abiotic stresses. They are important signal molecules that can participate in the regulation of plant defense signal transduction pathway and plant defense gene expression [49,50]. Several studies have shown that plant hormone are involved in the NBS-LRR protein-mediated defense responses [51].

ABA is a positive or negative regulator in the process of plant defense against various pathogens [52,53]. Exogenous application of ABA enhances rice resistance to *Cochliobolus miyabeanus* [54]. In addition, the application of ABA increases the susceptibility of *Arabidopsis* plants to *Pseudomonas syringae* [55]. JA helps to regulate the plant defense response to insect and microbial pathogens [56]. Niu [57] pointed out that JA was a signal molecule involved in wheat powdery mildew resistance. Exogenous application of MeJA to *Arabidopsis thaliana* helps protect it against gray mold [58,59]. SA is vital in plant defense; it is one of the key signaling molecules involved in activating pathogen resistance [60,61]. The *Arabidopsis* NBS-LRR gene *RRS1-R*-mediated resistance to *Ralstonia solanacearum* depends on SA. Plant infection by pathogens can promote ET production and in turn induce the expression of a series of genes related to the disease course [62,63]. ET signaling also interacts with JA and SA signaling pathways to help plants resist the invasion of pathogenic bacteria [64,65,66,67]. The *RCY1* gene, which encodes the *Arabidopsis* CC-NBS-LRR protein to confer resistance to the cucumber mosaic virus, requires SA and ET signal transduction mechanisms [68]. GA promotes plant growth by stimulating the degradation of the negative growth regulator DELLA protein. DELLA protein controls the plant’s immune response by regulating the balance of JA and SA signals, indicating that GA is essential in response to pathogens’ invasion [69]. Reactive oxygen species (ROS) production is excessive in the early stages of plant–pathogen interactions. Appropriate levels of ROS can not only promote cell wall enhancement and plant antitoxin production but also have a signaling role in plant defense responses [70]. ROS signaling is involved in the resistance of transgenic cotton (expressing endochitinase gene) to *Rhizoctonia solani* [71]. The wheat NBS-LRR gene *TaRCR1* maintains ROS’s stability by regulating the transcription of ROS-scavenging enzyme gene *CAT1* and peroxidase gene *POX2*. *TaRCR1* then regulates the expression of defense-related genes to enhance the resistance to *Rhizoctonia cerealis* [70]. In this study, cucumber leaves were sprayed with different exogenous substances, such as ABA, MeJA, SA, ETH, GA, and H_2_O_2_. The qRT–PCR analysis showed that both *CsRSF1* and *CsRSF2* genes could be induced and expressed to different degrees. These results suggested that *CsRSF1* and *CsRSF2* may be involved in the crosstalk of exogenous substances against *S. fuliginea*. In addition, the expression of the *CsRSF1* gene was upregulated and then downregulated at all treatment time points and reached the peak 12 h after ABA treatment. After GA treatment, the expression of the *CsRSF2* gene was downregulated at all treatment time points. The results indicated that *CsRSF1* might be associated with the ABA signaling pathway in defense responses. *CsRSF2* might be involved in defense responses of the GA signaling pathway. However, their mechanism of action is currently unclear.

The rice NBS-LRR resistance protein Pit interacts with the GEF OsSPK1 to activate OsRac1 and trigger rice immunity [72]. The *Gossypium hirsutum* NBS-LRR gene *GhDSC1* modulates defense responses of ROS accumulation and expression of JA-regulated defense genes [73]. By now, the regulatory mechanisms of NBS-LRR gene resistance to powdery mildew and it activates the defense response by which signaling pathway are unclear, especially in cucumber. Our study lay a foundation for exploring regulatory mechanisms of NBS-LRR gene resistance to powdery mildew in cucumber.

## 4. Materials and Methods

### 4.1. Plant Materials and Treatments

Cucumber B21-a-2-2-2 (highly susceptible to *S. fuliginea*), B21-a-2-1-2 (highly resistant to *S. fuliginea*) and XinTaiMiCi were used in this study for gene transformation. B21-a-2-2-2 and B21-a-2-1-2 are two sister lines. Cucumber plants were grown in a room with a day/night temperature of 25 °C/22 °C and a 16 h light photoperiod.

### 4.2. Pathogen Growth and Inoculation

*S. fuliginea* was collected from the naturally infected cucumber plants of the Vegetable Institute of the Liaoning Academy of Agricultural Sciences (Shenyang, China). The pathogen was cultured on leaves of potted cucumber and amplified at 25 °C in the laboratory. Spores of *S. fuliginea* were harvested from highly infected cucumber leaves by washing leaves with sterile water containing 10 μg·mL^−1^ SDS and filtering with sterile gauze. The spore concentration was adjusted to 30–40 spores per field of vision (10 × 10 times under microscope). Then the spore suspension was sprayed evenly on cucumber leaves without *S. fuliginea* using a throat sprayer [74].

### 4.3. Full-Length CDS Cloning and Sequence Analysis

Primers (F and R) were designed according to the sequences of the target genes found on the Phytozome database (https://phytozome.jgi.doe.gov/pz/portal.html, accessed on 16 July 2018). Primer design was conducted using Primer 5 software (Premier, Mississauga, Ontario, Canada). The gene-specific primers are listed in Appendix A. Total RNA was extracted from the leaves of B21-a-2-1-2 and B21-a-2-2-2 using an RNAprep pure plant kit (Tiangen Biotech, Beijing, China) and synthesized into cDNA using the FastQuant cDNA (Tiangen Biotech, Beijing, China). The following PCR program was used: 98 °C (10 s), 56 °C (15 s), and 68 °C (3 min) for 35 cycles. The reaction was conducted in a 20 µL volume containing 5 µL of 5× PrimeSTAR GXL buffer, 2 µL of dNTP mixture, 0.5 µL of each primer, 2 µL of genomic DNA, 0.5 µL of PrimeSTAR GXL DNA polymerase, and 8.5 µL of doubly distilled water. DNAMAN software (Lynnon Biosoft, San Ramon, CA, USA) was used for sequence alignment. Bioinformatics software was used for sequence analysis.

### 4.4. Subcellular Localization

The coding regions of *CsRSF1* and *CsRSF2* were amplified using PrimeSTAR GXL DNA polymerase (Takara, Dalian, China). The PCR amplification procedure was as follows: 98 °C (10 s), 56 °C (15 s), and 68 °C (3 min) for 35 cycles. The PCR products were detected by agarose gel electrophoresis and ligated to the 3′ end of the GFP coding region without the stop codon in the p35S:GFP vector, generating the fusion constructs p35S:GFP-*CsRSF1* and p35S:GFP-*CsRSF2*. The recombinant plasmids p35S:GFP-*CsRSF1*, p35S:GFP-*CsRSF2*, and p35S:GFP were separately transformed into *Agrobacterium tumefaciens* strain EHA105 and then centrifuged after overnight culture. The precipitate after centrifugation was cultured in induction medium (10 mmol·L^−1^ 2-(4-morpholino) ethanesulfonic acid (MES), pH 5.7, 10 mmol·L^−1^ MgCl2 and 200 mmol·L^−1^ acetosyringone (AS)), collected and diluted to OD_600_ = 0.6, and then infiltrated into *Nicotiana benthamiana* leaves. The plants were incubated at 25 °C for 3 days after injection, and point GFP signals were observed and photographed using a laser confocal fluorescence microscope (Leica, TSC SP8, Solms, Germany) with an excitation wavelength of 488 nm and 505 to 530 nm bandpass emission filter.

### 4.5. Abiotic and Biotic Stresses

1-month-old and 9-day-old cucumber seedlings (B21-a-2-1-2 and B21-a-2-2-2) were used in the following treatments. The leaves of 1-month-old cucumber seedlings were sprayed with 100 µmol·L^−1^ ABA, 100 µmol·L^−1^ MeJA, 1 mmol·L^−1^ SA, 10 μL·L^−1^ ETH, 100 mg·mL^−1^ GA, and 10 μmol·L^−1^ H_2_O_2_ to simulate abiotic stress, while control seedlings were sprayed with distilled water. The cotyledons of 9-day-old cucumber seedlings were sprayed with *S. fuliginea* to simulate biotic stress, while control seedlings were sprayed with distilled water. The leaves of the abiotic stress/biotic stress–treated seedlings were harvested at indicated time points (12 h, 24 h, and 48 h and 0 h, 3 h, 6 h, 9 h, 12 h, and 24 h, respectively), frozen in liquid nitrogen, and then stored at −80 °C until further use.

### 4.6. Tobacco Rattle Virus (TRV)-Mediated CsRSF1/CsRSF2 Gene Silencing

The pTRV2:*CsRSF1* and pTRV2:*CsRSF2* recombinant constructs were generated by subcloning a 233 bp-specific fragment within the 3′ region of the sequence of *CsRSF1* (from 2225 to 2457 nucleotides in *CsRSF1* CDS sequence) and a 289 bp-specific fragment within the 5′ region of the sequence of *CsRSF1* (from 1 to 289 nucleotides in *CsRSF1* CDS sequence) in the antisense orientation into the pTRV2. The gene-specific primers are listed in Appendix A. Then, the pTRV2:*CsRSF1* and pTRV2:*CsRSF2* recombinant constructs were introduced into different *A. tumefaciens* EHA105 aliquots. A 12.5 mL culture of different *A. tumefaciens* (pTRV2:*CsRSF1*, pTRV2:*CsRSF2*, and pTRV2, pTRV1) strains were grown overnight at 28 °C in the YEP medium supplemented with 50 mg·L^−1^ of rifampicin and 50 mg·L^−1^ of kanamycin. Then, each 100 µL overnight culture was inoculated into 12.5 mL portions of the YEP medium with the aforementioned antibiotics and cultivated at 28 °C until the culture reached selected optical densities of OD_600_ = 0.8–1.0. The samples were supplemented with 10 mmol·L^−1^ MES, 10 mmol·L^−1^ MgCl_2_, and 200 µmol·L^−1^ AS until the OD_600_ = 0.4 for each construct. The induced *A. tumefaciens* EHA105 strains carrying different pTRV2-derived vectors (pTRV2, pTRV2:*CsRSF1*, and pTRV2:*CsRSF2*) were mixed with the pTRV1 *A. tumefaciens* strain EHA105 in a ratio of 1:1. Then, the samples were co-infiltrated into fully expanded cotyledons of cucumber plants using a 1 mL syringe. The plants were placed in a room at 22 °C with a 16 h light and 8 h dark photoperiod for growth.

### 4.7. CsRSF1 and CsRSF2 Overexpression Transformation

The full CDS sequences of the *CsRSF1* and *CsRSF2* genes were subcloned into a modified pRI101-AN-GFP vector with a GFP tag, resulting in the transformation vector pRI101-AN-GFP:*CsRSF1* and pRI101-AN-GFP:*CsRSF2*. The cloning site was between *Bam*HI and *Sal*I. The gene-specific primers are listed in Appendix A. The constructed vectors and pRI101-AN-GFP were transformed into *A. tumefaciens* strain EHA105. The transformation method was the same as described earlier.

### 4.8. Quantitative Reverse-Transcription PCR (qRT–PCR)

Total RNA was extracted from cucumber euphylla, and cotyledons using an RNAprep pure plant kit (Tiangen Biotech, Beijing, China) and was used for cDNA synthesis using a fasting RT Kit cDNA (Tiangen Biotech, Beijing, China). The primers were designed using Primer 5 software (Premier, Mississauga, ON, Canada), and the PCR reaction quality was estimated based on melting curves. Three biological replicates of cDNA were used for three repeated experiments. The actin gene of cucumber was used as the internal reference. Quantitative reverse-transcription PCR (qRT–PCR) was used to detect the transcription level of *CsRSFs* under different treatments using the SYBR Green I 96-I system (Roche fluorescence quantitative PCR instrument, Basel, Switzerland). The reaction mixtures consisted of 4.5 μL of 2× SuperReal PreMix Plus (Tiangen Biotech, Beijing, China), 0.2 μL of primers (0.1 μL of the forward primer and 0.1 μL of reverse primer), 4.3 μL of RNase-free ddH_2_O, and 1 μL of cDNA. The PCR program was set up in seven stages: (1) 95 °C for 15 min (pre-incubation), (2) 95 °C for 10 s, (3) 58 °C for 20 s, (4) 72 °C for 30 s, (3) repeated 40 times (amplification), (5) 95 °C for 0.5 s, (6) 60 °C for 1 min and (melt), and (7) 50 °C for 30 s (cooling). The relative expression levels of the target genes were calculated using the 2^−ΔΔ*C*t^ method. The significance was determined by the Student’s *t*-test (*p* < 0.05 or *p* < 0.01) using SPSS statistical software (SPSS 22.0, Nlinedown, Guangdong, China). All the primers are listed in (Appendix A).

### 4.9. Extraction of Cucumber Protoplasts

The cucumber cotyledons transformed with recombinant vectors were cut into 0.5 to 1 mm thin strips, quickly transferred to the enzymatic hydrolysate [20 mmol·L^−1^ MES, 1.5% cellulase R-10, 0.4% macerozyme R-10, 20 mmol·L^−1^ KCl, 0.4 mol·L^−1^ mannitol, 10 mmol·L^−1^ CaCl_2_, and 0.1% bovine serum albumin (BSA), pH 5.7], and shaken at 50 rpm for 6 h after 30 min of vacuum infiltration. The enzymatic hydrolysate was diluted with an equal amount of W5 (0.2 mol·L^−1^ MES, 1.54 mol·L^−1^ NaCl, 1 mol·L^−1^ CaCl_2_, and 0.2 mol·L^−1^ KCl, pH 5.7) and filtered through a 200 mm nylon membrane. The filtered solution was centrifuged at 100× *g* and 4 °C for 2 min. The supernatant was gently removed with a pipette. The remaining green liquid, which contained protoplasts, was placed on the ice for 30 min. Finally, the protoplasts were resuspended in the same volume of pre-cooled MMG (0.2 mol·L^−1^ MES, 0.4 mol·L^−1^ mannitol, and 1.5 mol·L^−1^ MgCl_2_) and then observed under a laser fluorescence microscope (Leica, TSC SP8, Solms, Germany).

### 4.10. Coomassie Brilliant Blue R250 Staining and Microscopy

Coomassie brilliant blue R250 staining was used to observe the *S. fuliginea* hyphae on the cotyledons with the control of TRV:00, TRV:*CsRSF1*, TRV:*CsRSF2*, GFP: 00, GFP:*CsRSF1*, and GFP:*CsRSF2* transgenic plants on day 7 with *S. fuliginea*. The infected cotyledons of control and transgenic plants were immersed in a destaining solution (containing 0.225 g trichloroacetic acid, 150 mL of absolute ethanol, and 50 mL of acetone) for 30 min at 70 °C. The samples were then stained with Coomassie brilliant blue R250 solution (containing 0.225 g trichloroacetic acid, 0.9 g Coomassie brilliant blue R250, and 150 mL of methanol) for 2 min at room temperature. After the samples were washed with sterile water, they were immersed in 20% glycerol and photographed. The fungus was visualized under a light microscope (Nikon Ts2, Tokyo, Japan).

### 4.11. Disease Index

The typical symptoms of pathogen whiteness were measured in six disease severity ratings from 0 to 9, where 0 denoted no symptom, 1 denoted white lesions accounting for less than 5% of the entire inoculated leaves, 3 denoted white lesions accounting for 5%–25% of the entire inoculated leaves, 5 denoted white lesions accounting for 26%–50% of the entire inoculated leaves, 7 denoted white lesions accounting for 51%–75% of the entire inoculated leaves, and 9 denoted white lesions accounting for more than 75% of the entire inoculated leaves. Disease index (DI) was calculated using the following formula: DI (%) = [(*N* × *D*)/(*H* × *T*)] × 100, where *N* is the number of leaves with the respective disease rating, *D* is the disease rating, *H* is the highest disease rating, and *T* is the total number of observed leaves [75]. No fewer than 60 cucumber leaves were surveyed for the DI evaluation.

### 4.12. Data Processing and Statistical Analysis

Data were the mean ± standard deviation of three biological replicates per cultivar. The 2^−ΔΔ*C*t^ method was used to calculate the relative expression levels of the target genes. Standard errors of deviation were assessed using the STDEVA function in Excel. Statistical significance was analyzed using the Student’s *t*-test (*p* < 0.05 or *p* < 0.01) via SPSS software (SPSS 22.0, Nlinedown, Guangdong, China).

### 4.13. Bioinformatics Analysis Content and Tools Website

The primary structure was analyzed by using ProtParam. The website is http://web.expasy.org/protparam/, accessed on 9 April 2019 . The domain was analyzed by using SMART and InterProScan. The websites are http://smart.embl-heidelberg.de/, accessed on 9 April 2019 and ftp://ftp.ebi.ac.uk/pub/soft-ware/unix/iprscan/5/, accessed on 9 April 2019.

## 5. Conclusions

Cucumber *CsRSF1* and *CsRSF2* genes acted as positive modulators in the response of cucumber to *S. fuliginea* infections, and their regulatory mechanisms might be different. They participated in the defense response to *S. fuliginea* by regulating the expression of certain defense-related genes. In summary, this research provided new insights into the important roles of *CsRSF1* and *CsRSF2* in the resistance to *S. fuliginea.* This research laid a good foundation for exploring the molecular mechanism of cucumber resistance to *S. fuliginea* and breeding resistant varieties.

## Figures and Tables

**Figure 1 ijms-22-03986-f001:**
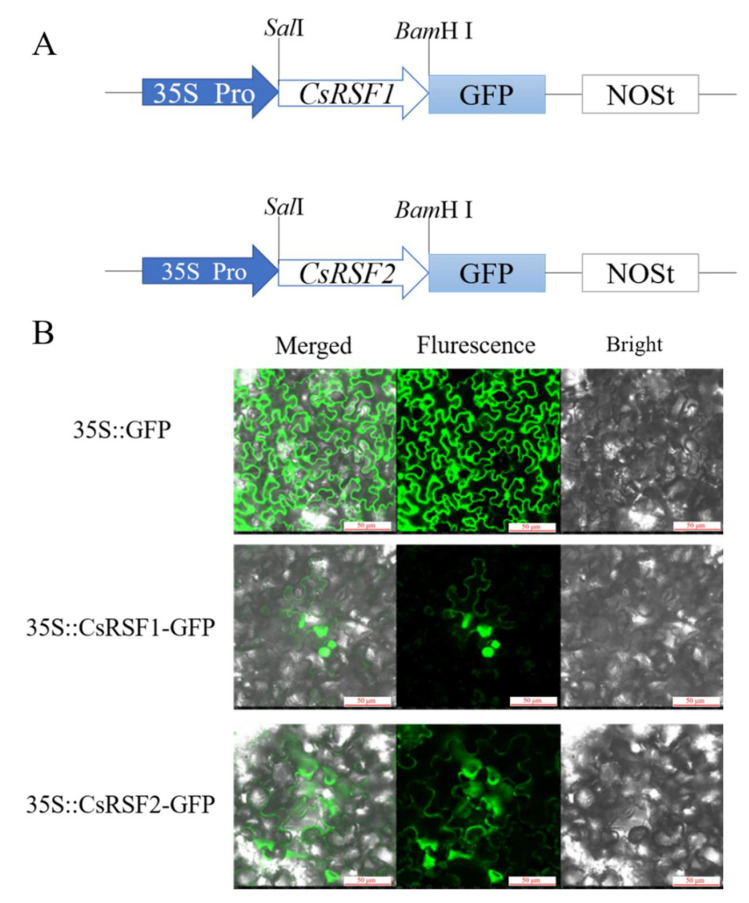
Subcellular localization of CsRSF1 and CsRSF2 in the *Nicotiana benthamiana* epidermis cell. (**A**) Schematic of p35S::*CsRSF1*-GFP and p35S::*CsRSF2*-GFP constructs used for the subcellular localization. (**B**) Green fluorescence protein (GFP) was detected in the *N. benthamiana* cell of the 35S::GFP, 35S::*CsRSF1*-GFP and 35S::*CsRSF2*-GFP constructs. Fluorescence, bright-field and merged images were obtained using a Leica confocal microscope. Bars = 50 µm.

**Figure 2 ijms-22-03986-f002:**
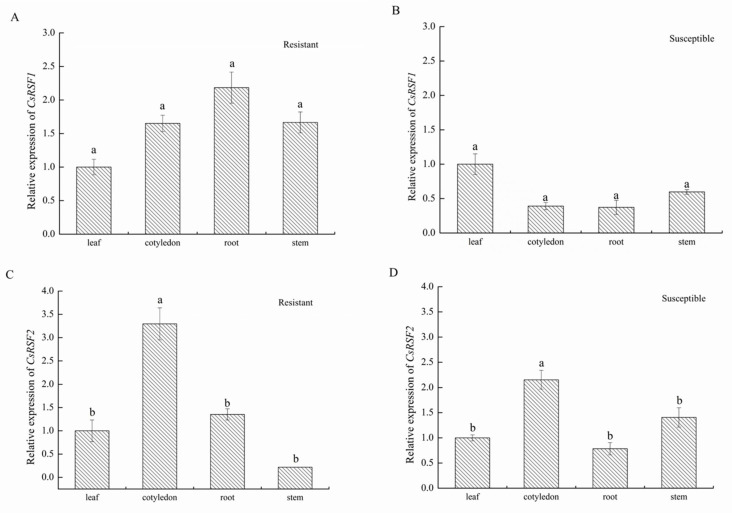
Expression patterns of *CsRSF1* and *CsRSF2* in various cucumber organs (leaf, cotyledon, stem and root). (**A**) Expression pattern of *CsRSF1* in resistant variety. (**B**) Expression pattern of *CsRSF1* in susceptible variety. (**C**) Expression pattern of *CsRSF2* in resistant variety. (**D**) Expression pattern of *CsRSF2* in susceptible variety. The expression level in the leaf was normalized as 1. Expression analysis of candidate genes using the 2^−ΔΔ*C*t^ method. Data are means ± SD from three biological replicates per cultivar. Different letters (a,b) above the bars indicate a significant difference determined by Student’s *t*-test (*p* < 0.05).

**Figure 3 ijms-22-03986-f003:**
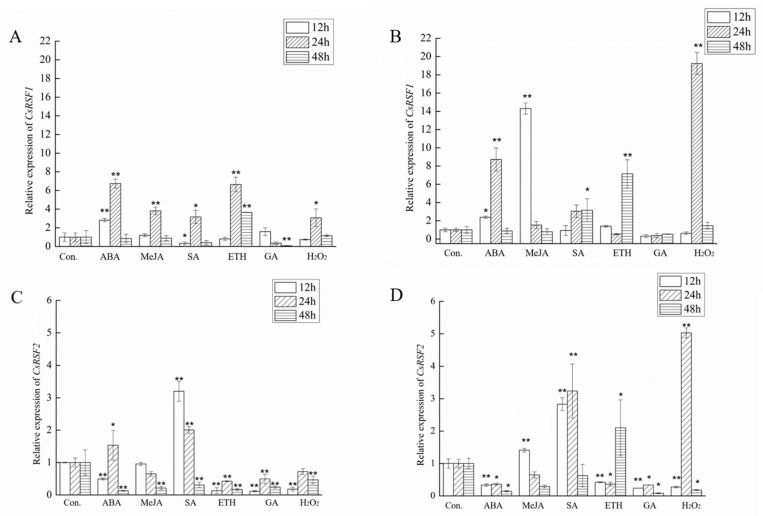
Expression patterns of *CsRSF1* and *CsRSF2* genes under exogenous substances. These exogenous substances include 100 µmol·L^−1^ abscisic acid (ABA), 100 µmol·L^−1^ jasmonate (MeJA), 1 mmol·L^−1^ salicylic acid (SA), 10 μL·L^−1^ ethephon (ETH), 100 mg·mL^−1^ gibberellin (GA), 10 μmol·L^−1^ hydrogen peroxide (H_2_O_2_). (**A**) qRT–PCR analyses of *CsRSF1* transcripts in the resistant cultivar. (**B**) qRT–PCR analyses of *CsRSF1* transcripts in the susceptible cultivar. (**C**) qRT–PCR analyses of *CsRSF2* transcripts in the resistant cultivar. (**D**) qRT–PCR analyses of *CsRSF2* transcripts in the susceptible cultivar. The relative expression levels of *CsRSF1* and *CsRSF2* in cucumber plants at various time points (12 h, 24 h, 48 h) were compared with the mock control, which was set to 1. Expression analysis of candidate genes using the 2^−ΔΔ*C*t^ method. Data are means ± standard deviations (SD) from three biological replicates per cultivar. The asterisks indicated a significant difference (Student’s *t*-test, * *p* < 0.05 or ** *p* < 0.01).

**Figure 4 ijms-22-03986-f004:**
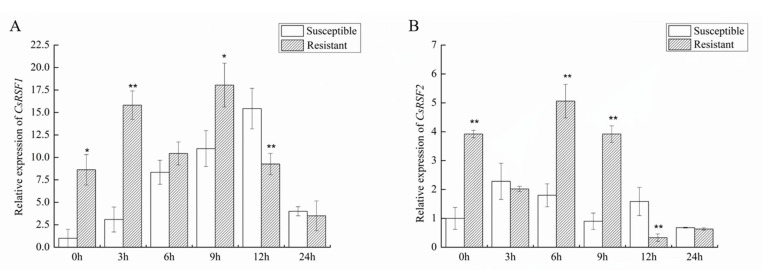
Expression patterns of *CsRSF1* and *CsRSF2* in resistant (B21-a-2-1-2) and susceptible (B21-a-2-2-2) varieties were inoculated with *S. fuliginea*. (**A**) The expression of *CsRSF1* in resistant and susceptible varieties were inoculated with *S. fuliginea*. (**B**) The expression of *CsRSF2* in resistant and susceptible varieties were inoculated with *S. fuliginea*. Expression analysis of candidate genes at 0, 3, 6, 9, 12 and 24 hpi (hours post-inoculation) using the 2^−ΔΔ*C*t^ method. The expression level in B21-a-2-2-2 was the mock control as 1. Data are means ± standard deviations (SD) from three biological replicates per cultivar. The asterisks indicated a significant difference (Student’s *t*-test, * *p* < 0.05 or ** *p* < 0.01).

**Figure 5 ijms-22-03986-f005:**
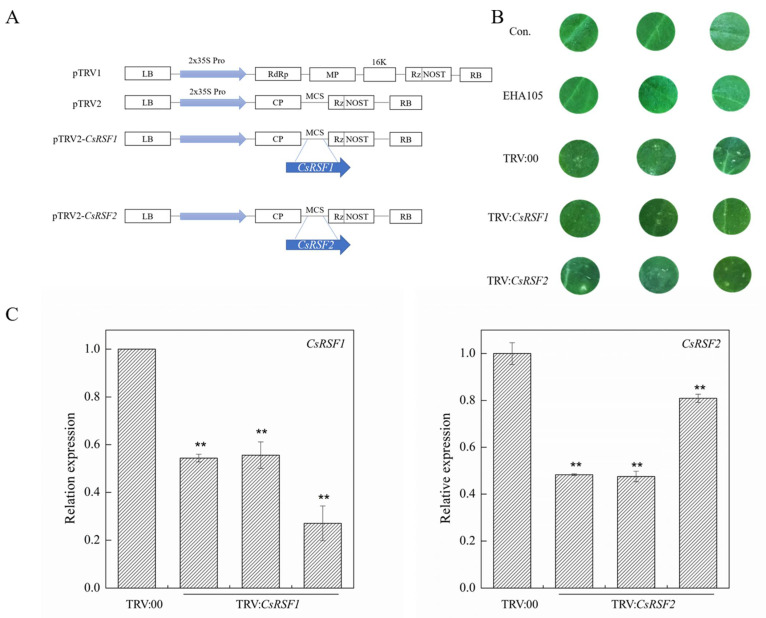
Identification of *CsRSF1*/*CsRSF2*-silencing cucumber plants. (**A**) Schematic of the *CsRSF1*/*CsRSF2*-silenced constructs. (**B**) Symptoms in detached cotyledons of silencing cucumber plants. Con. indicates non-injected plants. EHA105 indicates EHA105-injected plants. (**C**) *CsRSF1*/*CsRSF2*-silenced were identified in transgenic plants by qRT–PCR. TRV:00 indicates TRV: 00-injected plants, TRV: *CsRSF1*/*CsRSF2* indicates silencing plants. Data are means ± standard deviations (SD) from three independent experiments, and each column represents a sample containing three cucumber cotyledons from different plants. Expression analysis of candidate genes using the 2^−ΔΔ*C*t^ method. The asterisks indicated a significant difference (Student’s *t*-test, ** *p* < 0.01).

**Figure 6 ijms-22-03986-f006:**
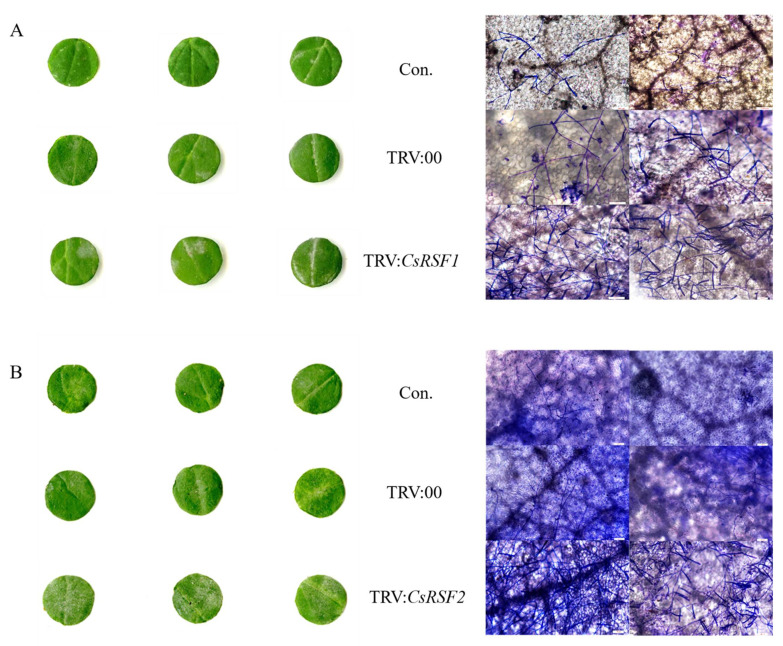
Identification disease-resistance of *CsRSF1*/*CsRSF2*- silencing plants after *S. fuliginea* infection. (**A**) Phenotypic analysis and Coomassie brilliant blue staining to examine *CsRSF1* transient silencing cucumber cotyledons after *S. fuliginea* infection. (**B**) Phenotypic analysis and Coomassie brilliant blue staining to examine *CsRSF2* transient silencing cucumber cotyledons after *S. fuliginea* infection. Con. indicates non-injected plants, TRV:00 indicates TRV:00-injected plants, TRV: *CsRSF1*/*CsRSF2* indicates silencing plants. Bars = 100 µm.

**Figure 7 ijms-22-03986-f007:**
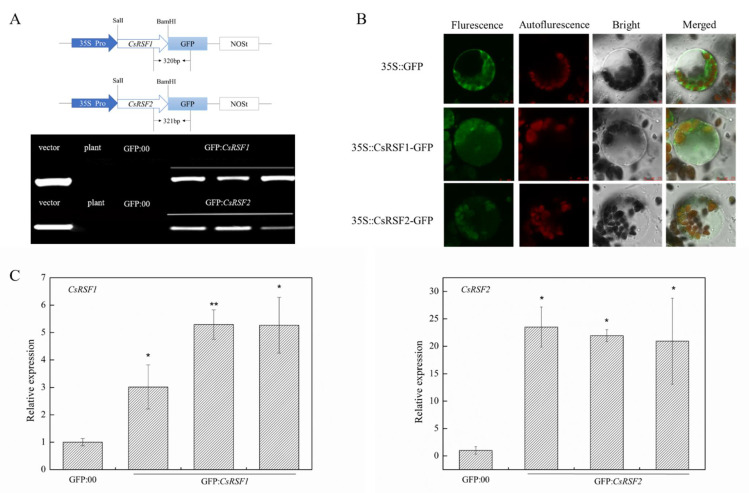
Identification of *CsRSF1*/*CsRSF2*-overexpressing cucumber plants. (**A**) Schematic of the *CsRSF1*/*CsRSF2*-GFP constructs. *CsRSF1* and *CsRSF2* were between 35S Pro and GFP protein, and chimeric PCR identifications of *CsRSF1* and *CsRSF2* genetically modified cucumber were successful. Vector, recombinant plasmid; plant, non-transgenic cucumber; GFP: 00, empty vector; GFP:*CsRSF1*, *CsRSF1*-transient overexpressing in cucumbers; GFP:*CsRSF2*, *CsRSF2*-transient overexpressing in cucumbers. (**B**) Luminescence signal identification for transient overexpressing cucumber cotyledons. Bar = 10 µm. (**C**) *CsRSF1*/*CsRSF2*-overexpressing were identified in transgenic plants by qRT–PCR. Data are means ± standard deviations (SD) from three independent experiments, and each column represents a sample containing three cucumber cotyledons from different plants. Expression analysis of candidate genes using the 2^−ΔΔ*C*t^ method. The asterisks indicated a significant difference (Student’s *t*-test, * *p* < 0.05 or ** *p* < 0.01).

**Figure 8 ijms-22-03986-f008:**
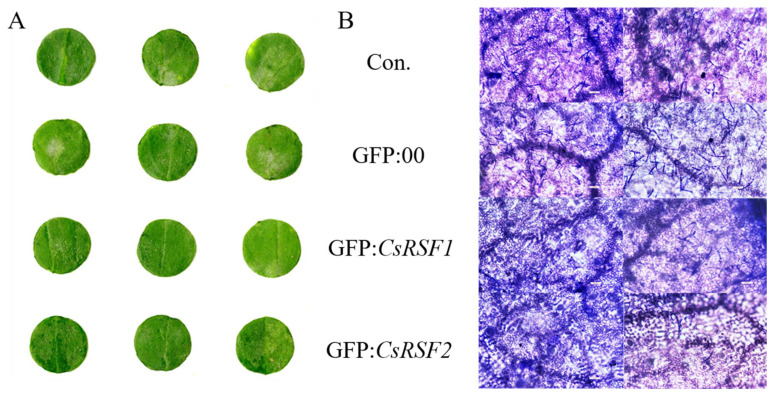
Identification disease-resistance of *CsRSF1*/*CsRSF2*- overexpressing plants after *S. fuliginea* infection. (**A**) Phenotypic analysis for *CsRSF1* and *CsRSF2* transient overexpressing cucumber cotyledons after *S. fuliginea* infection. (**B**) Coomassie brilliant blue staining for examining *CsRSF1* and *CsRSF2* transient overexpressing cucumber cotyledons after *S. fuliginea* infection. Con. indicates non-injected plants, GFP:00 indicates GFP:00-injected plants, GFP: *CsRSF1*/*CsRSF2* indicates overexpressing plants. Bars = 100 µm.

**Figure 9 ijms-22-03986-f009:**
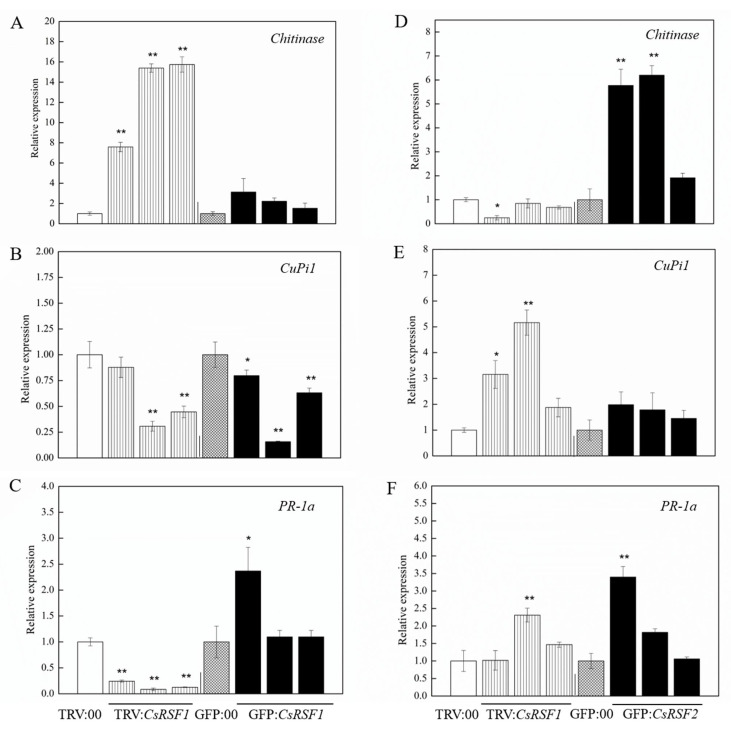
*CsRSF1* and *CsRSF2*-mediated expression patterns of defense-related genes (*Chitinase*, *CuPi1*, *PR-1a*) in transient silencing (TRV) and overexpressing (GFP) cucumber cotyledons. (**A**–**C**) The relative expression of *Chitinase*, *CuPi1*, *PR-1a* in *CsRSF1* -transient silencing and overexpressing cucumber cotyledons. (**D**–**F**) The relative expression of *Chitinase*, *CuPi1*, *PR-1a* in *CsRSF2* -transient silencing and overexpressing cucumber cotyledons. Data are means ± standard (SD) deviations from three independent experiments, and each column represents a sample containing three cucumber cotyledons from different plants. Expression analysis of candidate genes using the 2^−ΔΔ*C*t^ method. The asterisks indicated a significant difference (Student’s *t*-test, * *p* < 0.05 or ** *p* < 0.01).

**Table 1 ijms-22-03986-t001:** Disease index investigation for *CsRSF1* transient silencing cucumber cotyledons after *S. fuliginea* infection.

Material Name	Days Post-Inoculation (dpi)	Disease Index
Con.	7	22.22
TRV: 00	7	24.44
TRV:*CsRSF1*	7	32.41

**Table 2 ijms-22-03986-t002:** Disease index investigation for *CsRSF2* transient silencing cucumber cotyledons after *S. fuliginea* infection.

Material Name	Days Post-Inoculation (dpi)	Disease Index
Con.	7	16.68
TRV: 00	7	17.77
TRV:*CsRSF2*	7	36.11

**Table 3 ijms-22-03986-t003:** Disease index investigation for *CsRSF1* and *CsRSF2* transient overexpressing cucumber cotyledons after *S. fuliginea* infection.

Material Name	Days Post-Inoculation (dpi)	Disease Index
Con.	7	34.59
GFP: 00	7	32.00
GFP:*CsRSF1*	7	24.88
GFP:*CsRSF2*	7	17.33

## Data Availability

All data presented in article and Appendix A.

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
