# Peer review of "Nucleotide-Binding Leucine-Rich Repeat Genes *CsRSF1* and *CsRSF2* Are Positive Modulators in the *Cucumis sativus* Defense Response to *Sphaerotheca fuliginea"

_ijms, 2021, doi:10.3390/ijms22083986_

Round 1
Reviewer 1 Report
This paper describes the characterization and functional analysis of two NBS-LRR encoding genes CsRSF1 and CsRSF2 against powdery mildew infection in cucumber. The authors measure gene expression and infection rates in whole plants and different tissues in both susceptible/resistant lines as well as CsRSF1-2 knock-down and over-expressing plants treated with various growth and stress hormones. The authors find that these proteins both function in the cytoplasm and membrane as well as the nucleus (only CsRSF1). Knock-down (KD) CsRSF1/2 plants were more susceptible to powdery mildew while over-expressing (OE) lines were generally more resistant, suggesting a role in RSF1/2 genes in defense. These genes possibly function against infection by regulating the expression of various defense genes like chitinase and PR1, whose expression is also altered in KD and OE lines.
Overall, the paper follows a logical sequence of experiments, but there are several flaws that need to be addressed before publication. My comments are listed below:
- On lines 70-71 the authors state that there were no SNPs or DNA differences between the CsRSF1/2 genes in the resistant and susceptible plants. The authors find that the genes are differentially expressed in various tissues in these 2 lines (Supplementary figure S5). This suggests that there is significant epigenetic regulation of these genes and this may be conferring resistance or susceptibility. However, there is no discussion or mention of possible epigenetic control in the manuscript. Are there other known genes in other species that control the expression of RSF1/2? How could their differential expression in various tissues play a role? I am not sure why this important result was put into the supplementary materials.
- Figures 4A/B, 5A, 7A showing all the small circles is difficult to see in the paper and should be moved to the supplementary materials and enlarged.
- Figure 2: This would be easier to understand if each hormone were on its own graph and if resistant/susceptible lines were compared side by side. Or, the y-axis graphs should have the same scale so that the reader can compare data points across multiple graphs.
- Rename the legend in Figure 3 to Resistant/Susceptible rather than B21-a-2-2-2, etc.
- The authors use incorrect statistics for analyzing all of their data. Student t-tests should only be used when comparing 2 data sets. In the manuscript, multiple treatments/times/plant lines are being compared, including gene expression. Expression data is reported as a ratio and therefore needs a test for normality before analysis. Additionally, the authors should use either Bonferroni corrections or Kruskall-Wallis tests with ANOVA that is appropriate for these types of data.
- Figure 4 suggests that your silenced line is a knock-down and not a knock-out line.
- Figure 8: The graphs are not labeled A, B, C etc.
- Tables 1-3 can be combined into one table by adding a column for which treatment is being reported.
- The discussion is the weakest part of the manuscript and needs to be rewritten. The first paragraph (lines 239-250) are superfluous. The remainder of the discussion merely describes the roles of RSF1/2 in other species, but does not critically analyze or interpret the results of this paper or put them into a larger context. The authors could try to put together a model based on other species and their results which could biologically explain how RSFs might be playing a role in resistance to powdery mildew. Only stating that they may have a role in hormone signaling is not significant enough to warrant publication.
Reviewer 2 Report
Dear Authors,
I have an honor to review manuscript entiled „Nucleotide-Binding Leucine-Rich Repeat Genes CsRSF1 and CsRSF2 Are Positive Modulators in the Cucumis sativus Defense Response to Sphaerotheca fuliginea „ submitted to International Journal of Molecular Sciences MDPI.
Presented results provided some insights into the important roles of CsRSF1 and CsRSF2 in the resistance to S. fuliginea. This research can be a good background in exploring the molecular mechanism of cucumber resistance to cucmber powdery mildew.
Intresting and promising findings, presenting by Authors, concentrated on expression the transcript levels of CsRSF1 and CsRSF2, which were correlated with plant defense response against S. fuluginea. On the other hand, the silencing of CsRSF1 and CsRSF2 impaired host resistance to S. fuliginea, whereas CsRSF1 and CsRSF2 overexpression improved resistance to S. fuliginea in cucumber.
Paper is well and clear written, nevertheless I suggest some specific comments to the Authors:
- Please explain precisely , how was estimated „white lesions accounting in %” in disease index ?
- Please describe the aim of the study in manuscript, so precisely as it stays in the abstract part.
- „The results showed different tissue-specific (?)expression patterns of susceptible and resistant cultivars.” – tissue specific – I am sure that authors presented organ-specific or organ difference in genes expression (supplementary material); „various cucumber tissues (leaves, cotyledons, stems and roots).” ??? there are not plant’s tissues ,but definately plant organs.
- Bioinformatics analysis content and tools website – should be in materials and methods not in supplementary materials according journal rules.
- Figure 1 – it is not subcellular localisation – it is localisation in epidermis;
- In figure 2 – I suggest to make panels susceptible/resistance it will be more clear for the reader instead of finding what is A ,B or C, D;
- Figure 4B and 5 A and B – effect on leaflets are almost invisible – please enlarge it with a good resolution or change to a little bit bigger leaf areas in the photos;
- Figure 5- in Comassie staining the same situation- please make the section bigger to better find the differences; Figure 7 A and B the same situation;
- M&M section – from cucmber euphylla ??
Round 2
Reviewer 2 Report
Authors improved most, but not all my suggestions were taking into account;
In Figure 5B - 'small circle photos still stay' - the differences are still difficult to recognize
For example, we still have substantive error - As I have told before, plant leaves, cotyledons, stems there are plant organs, not tissues. Moreover, it is not explanation for the reviewer that "other/many people used it".
